# The Impact of Alcohol Consumption and Oral Microbiota on Upper Aerodigestive Tract Carcinomas: A Pilot Study

**DOI:** 10.3390/antiox12061233

**Published:** 2023-06-07

**Authors:** Marco Fiore, Antonio Minni, Luca Cavalcanti, Giammarco Raponi, Gianluca Puggioni, Alessandro Mattia, Sara Gariglio, Andrea Colizza, Piero Giuseppe Meliante, Federica Zoccali, Luigi Tarani, Christian Barbato, Marco Lucarelli, Flavio Maria Ceci, Silvia Francati, Giampiero Ferraguti, Mauro Ceccanti, Carla Petrella

**Affiliations:** 1Institute of Biochemistry and Cell Biology (IBBC-CNR), Sapienza University Hospital Policlinico Umberto I, 00161 Roma, Italy; marco.fiore@cnr.it (M.F.); christian.barbato@cnr.it (C.B.); 2Department of Sensory Organs, Sapienza University of Rome, 00161 Roma, Italy; antonio.minni@uniroma1.it (A.M.); luca.cavalcanti@uniroma1.it (L.C.); andrea.colizza@uniroma1.it (A.C.); pierogiuseppe.meliante@uniroma1.it (P.G.M.); federica.zoccali@uniroma1.it (F.Z.); 3Division of Otolaryngology-Head and Neck Surgery, San Camillo de Lellis Hospital, ASL Rieti-Sapienza University, Viale Kennedy, 02100 Rieti, Italy; 4Laboratory for Clinical Microbiology, Sapienza University Hospital Policlinico Umberto I, 00161 Roma, Italy; giammarco.raponi@uniroma1.it (G.R.); gianluca.puggioni@uniroma1.it (G.P.); 5Dipartimento della Pubblica Sicurezza, Direzione Centrale di Sanità, Centro di Ricerche e Laboratorio di Tossicologia Forense, Ministero dell’Interno, 00185 Roma, Italy; alessandro.mattia@poliziadistato.it; 6DIFAR—Department of Pharmacy, Università di Genova, Viale Cembrano 4, 16148 Genova, Italy; sara.gariglio@edu.unige.it; 7Department of Maternal Infantile and Urological Sciences, Sapienza University of Rome, 00161 Roma, Italy; luigi.tarani@uniroma1.it; 8Department of Experimental Medicine, Sapienza University of Rome, 00161 Roma, Italy; marco.lucarelli@uniroma1.it (M.L.); flaviomaria.ceci@uniroma1.it (F.M.C.); silvia.francati@outlook.it (S.F.); giampiero.ferraguti@uniroma1.it (G.F.); 9Pasteur Institute Cenci Bolognetti Foundation, Sapienza University of Rome, 00161 Roma, Italy; 10ASL Roma1, SITAC, Società Italiana per il Trattamento dell’Alcolismo e le sue Complicanze, 00100 Roma, Italy; mauro.ceccanti@uniroma1.it

**Keywords:** upper aero-digestive tract cancer, head and neck cancer, alcohol, oral microbiota, ADH polymorphism, oxidative stress, ROS

## Abstract

Alcohol consumption is associated with oxidative stress and an increased risk of carcinoma of the upper aero-digestive tract (UADT). Recently, it has been found that some microorganisms in the human oral cavity may locally metabolize ethanol, forming acetaldehyde, a carcinogenic metabolite of alcohol. In a cohort of patients first visited for UADT cancers, we estimated their alcohol consumption by measuring Ethyl Glucuronide/EtG (a long-lasting metabolite of ethanol) in the hair and carbohydrate-deficient transferrin/CDT (short-term index of alcohol intake) in the serum. Moreover, we analyzed, by culture-based methods, the presence of *Neisseria subflava*, *Streptococcus mitis*, *Candida albicans*, and *glabrata* (microorganisms generating acetaldehyde) in the oral cavity. According to the EtG values, we correlated drinking alcohol with endogenous oxidative stress and the investigated microorganism’s presence. We found that 55% of heavy drinkers presented microorganisms generating acetaldehyde locally. Moreover, we found that the presence of oral acetaldehyde-producing bacteria correlates with increased oxidative stress compared to patients without such bacteria. As for the study of alcohol dehydrogenase gene polymorphisms (the enzyme that transforms alcohol to acetaldehyde), we found that only the “CGTCGTCCC” haplotype was more frequent in the general population than in carcinoma patients. This pilot study suggests the importance of estimating alcohol consumption (EtG), the presence of bacteria producing acetaldehyde, and oxidative stress as risk factors for the onset of oral carcinomas.

## 1. Introduction

Alcoholic beverage consumption is strongly associated with an increased risk of certain cancers, particularly malignancies of the upper aerodigestive tract (UADT) [1,2,3,4]. Although the mechanisms underlying the effect of alcoholic beverage consumption on cancer risk are not fully understood, it is widely understood that acetaldehyde, the first metabolite of alcohol, may play an important role [5]. Acetaldehyde is a mutagen in animal models and is currently classified as a suspected human carcinogen [6,7]. Indeed, there is strong evidence for the carcinogenicity of acetaldehyde since it causes point mutations in human lymphocytes and induces sister chromatid exchanges and cross-chromosome aberrations. Moreover, acetaldehyde even interferes with the DNA repair machinery. 

Acetaldehyde is formed from ethanol by the action of the enzyme alcohol dehydrogenase (ADH) and is subsequently converted to acetate by aldehyde dehydrogenase (ALDH), primarily in the liver [8]. Alcohol metabolism has been associated with oxidative stress potentiation [9]. Indeed, oxidative stress may be considered an imbalance between the production of free radicals and the antioxidant defense system [10]. The toxic outcomes of ethanol are also regulated by oxidative stress through several mechanisms, such as the generation of crosslinks, lipid peroxidation, oxidative damage, DNA adducts, and DNA strand breaks [11]. At the cellular level, mitochondria are the primary source of reactive oxygen species (ROS) production. Elevated oxidative stress disrupts all the main biomolecules and induces modifications in many crucial cellular functions [12]. The effects that are particularly detrimental to the correct functioning of the alcohol target cells include altered cell signaling, mitochondrial dysfunction, cell growth inhibition, mutagenesis, and cancer process onset [13].

Recently, it has been found that some microorganisms (not only bacteria but also yeast fungi) in the human oral cavity are capable of locally metabolizing ethanol to form acetaldehyde [14,15,16]. Among the bacterial species identified in the human oral cavity, the genus *Neisseria* exhibited extremely high ADH activity and produced significant amounts of acetaldehyde when cultured in vitro in an ethanol-containing medium [17]. In addition, alcohol ingestion affects the bacterial composition of the oral microflora, resulting in increased levels of *Neisseria* [17]. Indeed, *Neisseria* spp. bacteria that are normally residents in human oral microflora are generally non-pathogenic; they could be a regional source of carcinogenic acetaldehyde and thus play a subtle role in alcohol-related carcinogenesis in human oral cancer [17].

The production of acetaldehyde from ethanol by oral streptococci has also been demonstrated in several studies [18,19,20,21,22]. Indeed, several streptococcal species, including *Streptococcus mitis*, have confirmed the metabolic property of acetaldehyde production from ethanol [18,19].

Finally, the production of carcinogenic acetaldehyde by *Candida* spp. has been associated with epithelial dysplasia and oral carcinogenesis [23,24,25]. *Candida albicans* isolated from patients with potentially malignant oral mucosal disorders can produce mutagenic levels of acetaldehyde. Further, concomitant cigarette smoking and alcohol consumption may promote changes in the oral microenvironment that lead to the upregulation of *Candida*-dependent acetaldehyde metabolism [24,26].

A growing number of studies have also shown that multiple ADH and ALDH genes are strongly associated with the onset, growth, and outcomes of UADT cancers [27,28,29]. In particular, changes in the polymorphisms of ADH1B, ADH7, ADH1C, and ALDH2 play fundamental roles in the pathogenesis of UADT malignancies [29,30,31].

Based on these findings, the aim of this pilot study was to analyze and correlate the following in a cohort of patients recruited for UADT carcinomas: (i) the long-lasting alcohol consumption levels by measuring ethyl glucuronide (EtG) (a metabolite of ethanol) in the hair and serum carbohydrate-deficient transferrin (CDT, a biomarker of alcohol consumption); (ii) the serum oxidative stress; (iii) the occurrence of *Neisseria subflava*, *Streptococcus mitis*, *Candida albicans*, and *glabrata* in the oral cavity; and (iv) the concomitant presence in the recruited individuals of alterations in ADH1B, ADH7, ADH1C, and ALDH2 gene polymorphisms [32]. 

## 2. Materials and Methods

### 2.1. Participants’ Selection and Sampling

In this study, we recruited 51 individuals suffering from UADT carcinomas, over a period of 2 years (2020–2021), who attended the Otolaryngology Clinic of the “Policlinico Umberto I”, Sapienza University Hospital. However, according to the inclusion/exclusion criteria described below, we enrolled 21 individuals in the study (17 adult men and 4 adult women).

Indeed, as inclusion criteria, we recruited adult male and female patients entering the hospital for the first diagnosis of UADT carcinomas. In terms of the exclusion criteria, we did not recruit patients who failed to provide informed consent; entered the hospital due to cancer relapse; were already undergoing specific drug treatments, such as chemotherapies, antidepressants, anti-inflammatory, and immunosuppressants; had received antibiotic/antifungal therapy up to 15 days prior to enrollment; and, finally, patients who were suffering from severe infectious diseases such as HIV, HBV, and HCV, other ongoing inflammatory, cardiovascular, endocrine, and autoimmune disorders, and had hair that was at least 5 cm long to acquire at least 5 months of information about their alcohol drinking (see EtG methods) [33].

Patients were enrolled at the time of diagnosis. All samplings were performed at the time of pre-hospitalization, within one week of diagnosis before the beginning of any kind of treatment. Peripheral blood samples of 5 mL were taken from each participant, collected in BD Vacutainer™ Serum Separation Tubes, and centrifuged at 3000 rpm for 15 min to separate serum from blood cells. Moreover, 5 mL of peripheral blood samples collected in BD Vacutainer™ with EDTA were collected for biochemical analysis and for RNA extraction. The serum and the blood were then stored at −80 °C up until the day of the analyses.

Non-tumor tissues were obtained from those patients who were eligible for surgical removal of the tumor (n. 11) and taken from the margins of the resection during intervention. These tissues were used for RNA extraction as controls for the analysis of gene expression of interest (see above).

The study was approved by the Sapienza University Hospital ethical committee (Rif. 6462); informed consent was signed by each patient and all the study procedures were in accordance with the Helsinki Declaration of 1975, as revised in 1983, for human experimentation.

### 2.2. Biochemical and Hematological Analyses

Differential blood cell count was performed using ADVIA 2120i Hematology System (Siemens Healthcare, Erlangen, Germany): MCV (males 80–96 fL; females 81–98 fL); HCT (males 40–52%; females 35–47%). Serum Fe (iron) (64.8–174.9 µg/dL) was analyzed using Cobas C 501 platform (Roche Diagnostics, Mannheim, Germany).

### 2.3. Ethyl Glucuronide (EtG) Measurement in the Hair

EtG is a metabolite of ethanol that is produced in the body by glucuronidation after drinking alcohol. It is used as an ethanol biomarker for checking and monitoring alcohol abstinence in circumstances where drinking is forbidden, such as by the military, liver transplant clinics, in professional monitoring programs (health professionals, attorneys, and airline pilots in recovery from addictions), in alcohol treatment programs, gestation, in schools, or in recovering alcohol-dependent individuals [34,35]. EtG is also utilized for checking the amount of alcohol use over time because it can be detected in hair and nails [36,37]. A substantial number of articles support the high sensitivity and specificity of hair EtG for the detection of heavy drinking [38,39]. 

Hair EtG analysis was performed according to the protocol described by Mattia et al. (2022) [40]. Briefly, 50 mg of hair samples were decontaminated, manually shredded, incubated overnight at 60 °C in water with 15 μL of EtG-D5 100 mg/L solutions in methanol, obtained from Medichem (GmbH & Co., Steinenbronn, Germany) as internal standard, purified with solid phase extraction polymeric cartridges Strata X-A-33 μm obtained from Phenomenex Inc. (Torrance, CA, USA), air-dried, and reconstituted with ACN and N-Methyl-N-(trimethylsilyl)-trifluoroacetamide (MSTFA) as derivatizing agent for injection in an Agilent Technologies 7890B gas chromatograph coupled to a 7000C tandem mass selective detector operating in EI ionization mode (Santa Clara, CA, USA). The GC separation was achieved by a Phenomenex Zebron 30 m × 250 μm × 0.25 μm column with a (5%-phenyl)-methylpolysiloxane stationary phase (Torrance, CA, USA). The injection temperature was set to 220 °C, and the injection volume was 2 μL. A pulsed splitless injection was used at 20 psi for 0.75 min. The solvent delay was about 3 min. The oven temperature was programmed as follows: isothermal at 100 °C for 1 min, then ramped at 30 °C/min up to 200 °C, held for 0 min, ramped again at 15 °C/min to 290 °C, and finally isothermal at 290 °C for 3 min. The total chromatographic run adds up to 13.3 min. The transfer line was held at 280 °C and the EI source at 230 °C.

EtG is expressed as pg/mg hair. EtG values > 30 pg/mg indicate alcohol abusers (referred to in the text and in the figures as “heavy drinkers”); 6–29 pg/mg values represent social drinkers, while ≤5 pg/mg values indicate non-drinkers [41].

### 2.4. Carbohydrate-Deficient Transferrin (CDT) Measurements

The determination of the CDT in the serum of the enrolled patients was carried out by capillary electrophoresis using CAPILLARYS 2 (Sebia Italia Srl). This test is based on the principle of capillary-free phase electrophoresis. The different glycoforms of transferrin, saturated in iron using a specific diluent, are separated inside silica capillaries according to their electrophoretic mobility under high voltage and alkaline buffer conditions. Glycoforms are detected directly by UV absorbance during migration. The migration takes place at a controlled temperature through a Peltier regulation device. The results are expressed as the sum of the percentages of disialo- and asialo-glycoforms with respect to the total transferrin. CDT in capillary electrophoresis (CE) values < 1.3% correspond to non-drinkers; 1.3% ≥ CDT in CE ≤ 1.6% values correspond to social drinkers; and CDT in CE values > 1.6% correspond to heavy drinkers.

### 2.5. Detection of Oral Microorganisms

The oral cavity swabs for the search of microorganisms were collected in the appropriate transport devices and promptly sent to the bacteriology laboratory for the analyses. Samples were cultured in elective and selective culture media (tryptocase soy agar, Columbia blood agar, Thayer–Martin agar, and CAF-Sabouraud dextrose agar, Becton Dickinson, Florence, Italy).

The identification of *Streptococcus mitis*, *Neisseria subflava*, *Candida albicans*, and *Candida glabrata* was performed on the microbial colonies grown after incubation at 37 °C for 48–72 in 5% CO_2_ atmosphere by a matrix-assisted laser desorption ionization time-of-flight mass spectrometry (MALDI-TOF MS) system (Bruker Daltonik GmbH, Bremen, Germany), with a discriminatory score > 2300. Any other growth of microorganisms was recorded and considered as “resident flora” when non-pathogenic, or “other” when potentially pathogenic microorganisms in the cultured clinical samples.

### 2.6. FORD (Free Oxygen Radicals Defense) and FORT (Free Oxygen Radicals Test) Analyses

As previously shown [42,43,44], by using kits (Callegari, Parma, Italy) for the analyses of ROS and the endogenous defense against free O_2_ radicals, we measured the serum oxidative stress status. Briefly, FORT is a colorimetric assay based on the ability of transition metals such as iron to catalyze, in the presence of hydroperoxides (ROOH), the formation of free radicals (reaction 1 and 2), which are then trapped by an amine derivative: CrNH_2_. The amine reacts with free radicals forming a colored, fairly long-living radical cation, detectable at 505 nm (reaction 3). The intensity of the color directly correlates to the number of radical compounds and the hydroperoxides concentration and, consequently, to the oxidative status of the sample according to the Lambert–Beer law.
1. R-OOH + Fe^2+^ → RO^•^ + OH^−^ + Fe^3+^
2. R-OOH + Fe^3+^ → ROO^•^ + H^+^ + Fe^2+^
3. RO^•^ + ROO^•^ + 2CrNH_2_ → ROO^−^ + RO^−^ + [CrNH^2•^]_purple_

As for FORD, this test uses preformed stable and colored radicals and determines the decrease in absorbance that is proportional to the blood antioxidant concentration of the sample according to Lambert–Beer’s law. In the presence of an acidic buffer (pH = 5.2) and a suitable oxidant (FeCl_3_), the chromogen (which contains 4-Amino-N,N-diethylaniline sulfate) forms a stable and colored radical cation that is photometrically detectable at 505 nm. Antioxidant compounds in the sample reduce the radical cation of the chromogen, quenching the color and producing a discoloration of the solution that is proportional to their concentration. The absorbance values obtained for the samples are compared with a standard curve obtained using Trolox (6-Hydroxy-2,5,7,8-tetramethylchroman-2-carboxylic acid), a permeable cell derivative of vitamin E that is commonly employed as an antioxidant.
1. Chromogen _(no color)_ + Fe^3+^ + H^+^ → Chromogen^•+^ _(purple)_
2. Chromogen^•+^ _(purple)_ + AOH → Chromogen+ _(no colour)_ + AO

### 2.7. ADH1B, ADH7, ADH1C, and ALDH2 Gene Polymorphisms

ADH1B, ADH7, ADH1C, and ALDH2 gene polymorphisms were carried out in the blood by minisequencing DNA analyses (ABI PRISM 3130xl Genetic Analyzer—Applied Biosystems—Waltham, MA, USA), while ADH7 and ALDH2 gene expressions were carried in the RNA of the available peritumoral tissues by digital droplet PCR (QX200 Droplet Reader and QuantaSoft software version 1.7.4—Bio-Rad. Hercules, CA, USA). DNA extraction was performed on 2 mL of whole blood collected in EDTA tubes, using QIAampDNA Mini Kit (Qiagen, Manchester, UK).

DNA was quantified by the Qubit fluorimetric assay (Invitrogen, Carlsbad, CA, USA). For RNA extraction, about 10 mg of normal, peritumoral tissue was obtained from each patient from the margins of resection. Peritumoral available tissue samples were immediately immerged in 1 mL of Trizol reagent and stored at −80 °C until the moment they were processed. The peritumoral available tissue was homogenized before extraction; RNA was obtained using Trizol Reagent (Invitrogen, Waltham, MA, USA) protocol, according to the manufacturer instructions. RNA from each available tissue was obtained in a volume of 20 µL. Extracted RNA was then quantified using Qubit fluorimetric assay (Invitrogen, Waltham, MA, USA). Detailed information is available in the Appendix A.

### 2.8. Statistical Analysis

Statistical analysis was performed using GraphPad Prism 5.01, a commercial scientific 2D graphing and statistics software (Boston, MA, USA). EtG findings were initially used to divide the enrolled individuals into 3 groups: non/very low-drinkers, social drinkers, and heavy drinkers. The presence or absence of mouth bacteria producing acetaldehyde and the severity of the discovered cancer (at both first hospitalization and the 9-month follow-up, according to the 8th edition of the AJCC/UICC—TNM staging system [45,46]) were also used as other fixed factors in the ANOVAs and MANOVAs of the analyzed clinical and biochemical parameters.

The sex effect was also considered, but since no relevant differences were found, this factor was removed. As for gene polymorphisms, we performed an initial descriptive analysis followed by a contingency analysis using the statistic tool. To analyze differences in categorical variables, we used the chi-square (χ^2^) test and Fisher’s exact test. To determine the statistical significance and the measure of significance testing, we calculated the probability value concept (*p*-value) with a confidence interval (CI) of 95%. A *p*-value of 0.05 was used as the cutoff for significance (statistically significant if *p* < 0.05).

## 3. Results

### 3.1. General Description of the Enrolled Individuals

The cohort of patients enrolled in the study (n. 21 with a middle age of 61.9 ± 1.7 years old) showed a higher prevalence of men compared to women (82% vs. 18%). Patients underwent a careful clinical diagnostic examination for the UADT carcinomas and were subsequently treated surgically. All the patients were declared smokers. The clinical (c) or pathological (p) tumor-node-metastasis staging system (TNM staging system, [45,46]) stadiation of each patient and 9 months of follow-up are shown in Table 1.

### 3.2. Ethyl Glucuronide (EtG) in the Hair

The analysis of the EtG in the hair provided data estimating the historical consumption (months) of alcohol. As shown in Figure 1, the amount of EtG showed that the percentage of heavy drinkers was 37% of the recruited patients (EtG > 30 pg/mL); the percentage of social consumers was 26%, while the percentage of non-drinkers’ patients was 37%.

As expected, first hospitalization blood routine analyses on mean corpuscular volume (MCV), iron (Fe), and hematocrit (HCT) confirmed and extended the EtG data by showing the highest values of these biomarkers in heavy drinkers compared to non-drinkers (F(2,18) = 12.33, 4.19, and 4.40, respectively, *p*s < 0.05 in the ANOVA, Figure 2).

### 3.3. Carbohydrate-Deficient Transferrin (CDT) Serum Levels

Another analysis of alcohol consumption was carried out by the examination of CDT (an ethanol biomarker available in the serum for up to 14 days) of the recruited patients. The results showed that, compared with EtG, there was an underestimation of heavy drinker individuals, which was probably due to a decrease in alcohol consumption before hospitalization.

Indeed, 80% of patients were not drinkers (CDT in capillary electrophoresis values < 1.3%), while 10% of patients were considered “heavy drinkers” (CDT in capillary electrophoresis values > 1.6 pg/mL). Another 10% of patients were “social consumers”, with CDT levels comprised between 1.3% ≥ CDT in capillary electrophoresis and ≤1.6% (Figure 3).

### 3.4. Cancer Localization Distribution

A total of 52% of the recruited patients had carcinomas in the larynx; 19% had cancer of the tongue; 14% had cancer in the hypopharynx; 10% in the oropharynx; and 5% in the upper gingiva (Figure 4).

According to both carcinoma localization and EtG values, non-drinkers showed a wider local distribution. In social drinkers, there was a prevalence of larynx tumor, while in heavy alcohol drinkers, carcinomas in the tongue, larynx, and hypopharynx were prevalent (Figure 5) [F(1,19) = 3.51, *p* = 0.05, Pillai test 0.28, in the MANOVA cancer district location x presence of acetaldehyde-producing bacteria x EtG].

### 3.5. Microbiota of the Oral Cavity

The analysis of the microbiota of the oral cavity was aimed at analyzing the oral swabs drawn from the enrolled patients in the presence of microorganisms with a high capacity to locally metabolize the alcohol to acetaldehyde, such as *Neisseria subflava*, *Streptococcus mitis*, *Candida albicans*, and *glabrata.* As shown in Figure 4, most of the non/very low drinkers or social drinkers had resident flora prevalence. *Candida glabrata* was observed in a small percentage (9%) of non-drinkers. Quite surprisingly, 55% of heavy drinkers displayed the presence microorganisms generating acetaldehyde locally in the oral cavity. In particular, 27% of patients presented *Streptococcus mitis*; 18% presented *Candida albicans*; and 10% presented *Neisseria subflava*. However, *Candida glabrata* was not disclosed in any of the enrolled heavy drinkers. Figure 6 shows the percentage distribution of the analyzed microorganisms.

### 3.6. Oxidative Stress Data

ANOVA data showed that drinking habits affected the oxidative stress of the recruited individuals (Figure 7, panels A and B). Indeed, the antioxidant capabilities expressed in the FORD data show that heavy drinkers (panel A) had the highest serum values, mainly when compared with the social drinkers [F(2,18) = 11.39, *p* < 0.01, see post hoc in the figure]. Furthermore, social drinkers had low serum levels of ROS measured in FORT (panel B), mainly when compared with the non-drinkers [F(2,18) = 3.35, *p* = 0.05, see post hoc in the figure].

As for the cancer severity (according to TMN) observed at both the first hospitalization and the 9-month follow-up, no differences in FORT and FORD were disclosed. Additionally, no differences in FORD and FORD were found according to cancer district location. However, the presence or absence of oral acetaldehyde-producing bacteria affected the oxidative stress of the enrolled individuals (Figure 7, panels A and B). In fact, the FORD and FORT levels were significantly elevated in cancer patients showing oral acetaldehyde-metabolizing bacteria [Fs(1,19) = 4.96 and 5.24, respectively, *p* < 0.05, see post hoc in Figure 8]. 

Based on FORT/FORD data, multivariate ANOVA analyses did not show significant interactions between alcohol drinking groups and cancer severity or location. Intriguingly, according to FORD and FORT MANOVA data, in the recruited individuals, the presence of oral acetaldehyde-producing bacteria correlated with the cancer district location [F(1,19) = 4.21, 5.30, *p*s < 0.05, Pillai Test 0.31, 0.37, respectively]. Indeed, patients with carcinoma in the larynx and hypopharynx disclosed a prevalence of resident oral flora, but patients with carcinoma in the upper district of the oral cavity showed a prevalence of acetaldehyde-producing bacteria (tongue, gingiva, and oropharynx).

### 3.7. ADH1B, ADH7, ADH1C, and ALDH2 Gene Polymorphisms and Haplotypes

We genotyped the single nucleotide polymorphisms (SNPs) listed above in the recruited patients. At the same time, 20 people from the general population, matched for age and sex, were genotyped and used as controls. We performed a statistical analysis of the allelic and genotypic frequencies. The analysis showed no significant differences between the patients and controls (Appendix A).

Using the FamHap software, we identified haplotypes with a frequency above 5% in the general population and in patients; we considered polymorphisms genotyped for ADH genes (respectively, rs1229984 and rs2066702 for ADH1B; rs1693482 for ADH1C; and rs894363, rs1154470, rs1154468, rs1573496, rs971074, and rs1154461 for ADH7 gene). The CGTCGTCCC haplotype was more frequent in the general population (25%) than in patients (6.8%) with a *p*-value of 0.0214 (Table 2). No differences between the alcohol habit groups based on EtG values were disclosed. No effects of the presence in the mouth of bacteria producing acetaldehyde were also evidenced.

### 3.8. ADH7 and ALDH2 Expressions in the Available Peritumoral Tissues

The available peritumoral tissues analysis of ADH7 expression showed that carcinoma patients with the haplotype “CGTCGTCCC” (*n* = four: two heavy drinkers, one social drinker, and one non-drinker) had a lower, but not significant (*p* = 0.09), copy number of ADH7 (33.82 copies/µL) with respect to cancer patients (*n* = 6) with other haplotypes (78.02 copies/µL). As for ALDH2, we found homogeneous expressions in the analyzed peritumoral tissues, and no differences emerged between patients carrying and those not carrying the “CGTCGTCCC” haplotype. Indeed, individuals with the “CGTCGTCCC” haplotype had an average of 1.50 copies/µL while patients with other haplotypes had an average of 1.68 copies/µL (Figure 9).

## 4. Discussion

In this study, we evaluated for the first time in a cohort of patients with a first diagnosis of UADT carcinoma whether or not long-lasting alcohol consumption could impact the localization of the tumor, the cancer severity and prognosis, and the alteration of the oral cavity microbiota as a function of the presence of some ADH and ALDH polymorphisms. In our study, all the patients were declared heavy smokers, which is also a common feature and risk factor for developing cancer in the upper aerodigestive tract. This fact allowed us to evaluate the potential contribution of alcohol consumption to the pathophysiology of cancer.

Firstly, we disclosed how the alcohol consumption of an individual will not always correspond to the declared one. Most of the scientific studies as well as clinical evaluations on tobacco and alcohol use are based on patient statements and generally do not involve objective evaluations [47,48,49]. Indeed, as for alcohol intake, the evaluation of the impact of alcohol consumption on carcinogenesis or the evolution of this kind of disease is not easily assessed. In our study, we provided an estimation of alcohol consumption both in short-term (CDT) and long-term periods before the study (EtG). Certainly, serum CDT estimates alcohol consumption over a period of a maximum of 15 days before the sampling. On the other hand, the determination of EtG in the hair allows us to assess the “historical” consumption of alcohol [40], which is a metabolite that irreversibly accumulates in the keratin matrix. In this way, taking into account the average hair growth of 1 cm per month, it is possible to evaluate the habits connected to the consumption of alcoholic beverages more accurately [33]. Interestingly, in our sample of the study, the percentage of heavy drinkers rose from 9% (according to CDT values) to 35% (according to EtG ranges), interesting data that sheds light on the significance of the potential role of alcohol in the carcinogenic processes of UADT. The data on routine blood analyses on MCV, Fe, and HCT confirmed and extended the EtG data by showing potentiated values of these biomarkers in heavy drinkers compared to non-drinkers.

Despite the evident prevalence of larynx carcinomas in the population considered, when we stratified according to alcohol consumption by EtG, we found that heavy drinker patients had tumors that were mainly confined to the tongue, while non-drinkers had a wider distribution. Risk factors for tongue cancer could include numerous factors, such as old age, geographical location, family history, nutritional deficiencies, infectious agents, and of course, chronic alcohol and tobacco abuse [50], even if the exact cause is unidentified. As for the relationship between alcohol consumption and the development of tongue cancer, this issue is not fully understood. A case–control study conducted in China [51] on the risk of tongue cancer associated with tobacco smoking and alcohol consumption revealed a non-association with alcohol consumption. However, a marginally significant association was found for those who consumed alcohol for a long period, suggesting its potential effect as a risk factor. Conversely, in another study [52], possible differences in risk factors between tongue cancer and tumors arising elsewhere in the oral cavity appear to be attributable to smoking, alcohol habits, and, dietary factors. Interestingly, there is a possibility that predisposing genetic factors may be a molecular basis for the onset of oral cavity cancer. In the past [53] but also more recently, genes associated with tongue cancer have been identified as differentially regulated in patients with a history of tobacco and/or alcohol use, concerning the non-habit group [54].

The incidence of alcohol-related diseases in Europe is increasing, particularly in Eastern Europe [55]. The best-known risks concern the consequences for the central nervous system, already observable as an acute effect linked to excessive alcohol consumption, and those affecting the liver (cirrhosis of the liver). Alcohol is also a type 1 carcinogen. In particular, today, alcohol is considered a risk factor for the development of neoplasms of the oral cavity, pharynx and larynx, esophagus, stomach, colorectal, liver, gallbladder, and pancreas [56,57,58,59]. According to recent results, people who limit themselves to consuming alcohol are 32 percent more likely to develop cancer of the mouth and throat compared to the general population. However, when the effects of smoking are added to this behavior, the risk almost increases tenfold [60].

To date, all the mechanisms through which alcohol contributes to the emergence of cancer are not known. However, it is known that alcohol, for example, irritates the mucous membranes, preventing damaged cells from repairing themselves properly [61]. This can promote the development of mouth and throat cancers. In addition to the direct action of alcohol, it is well known that its metabolism generates highly carcinogenic substances, such as acetaldehyde. As it is known, acetaldehyde is formed by the initial oxidation of ethanol in the liver by ADH; it is subsequently converted into acetic acid by ALDH. Quite interestingly, the evidence that the presence of some microorganisms in the oral cavity can favor the local formation of acetaldehyde may provide further information on the impact of alcohol consumption and the onset of UADT carcinomas.

Indeed, according to the results of the present experiment, we have shown that patients with carcinoma in the larynx and hypopharynx disclosed a prevalence of resident oral flora, but patients with carcinoma in the upper district of the mouth showed a prevalence of acetaldehyde-producing bacteria (tongue, gingiva, and oropharynx).

The role of the microbiota in maintaining the homeostasis of the organism has now been well defined [55]. The presence of alterations at this level (dysbiosis) may be associated with the onset of pathologies, or it can be an alarm bell for a morbid state. Even at the level of the oral cavity, the idea is that changes in the richness and diversity of the microbial ecosystem can be an indication, if not even the cause, of head and neck tumors [16,20]. Moreover, it is known that excessive alcohol consumption causes a modification in the microbiota of the oral cavity, reducing its diversity, favoring the impoverishment of advantageous species (e.g., lactobacilli), and, on the contrary, predisposing the district to the colonization of pathogenic microorganisms [16,62,63]. We therefore analyzed the presence of *Streptococcus mitis*, *Neisseria subflava*, and *Candida albicans* in the oral cavity. Although a small number of subjects were enrolled, it is evident that 55% of heavy drinkers presented at least one microorganism that was considered a strong metabolizer of alcohol by producing acetaldehyde in the oral cavity. Muto and co [17] demonstrated that among the bacterial species identified from the human oral cavity, genus *Neisseria* had extremely high ADH activity, producing significant amounts of acetaldehyde when cultured with a medium containing ethanol in vitro. *Neisseria spp.* in the oral microflora could be a significant regional source of carcinogenic acetaldehyde in response to the ingested ethanol and may potentially play an important role in alcohol-related carcinogenesis in human UADT. Bacterial acetaldehyde production from ethanol was also demonstrated in vitro [19]. In this study, the authors revealed that oral endogenous bacteria, *Streptococcus* and *Neisseria*, can produce acetaldehyde, depending on the environmental conditions (pH and aerobic and anaerobic conditions).

Concerning *Streptococcus* spp., the analysis of the oral microbiota in our enrolled patients revealed the presence of *S. mitis* in the oral cavity of heavy drinkers, supporting the hypothesis that its presence could participate in the regional formation of cancerogenic products. An interesting paper considering salivary microbiota as a diagnostic indicator of oral cancer demonstrated that *Streptococcus mitis* is one of the microorganisms whose presence is significantly associated with the presence of oral cancer [64]. Indeed, the authors found that the levels of salivary bacteria (i.e., *Streptococcus mitis*) in subjects with and without oral squamous cancer had higher diagnostic sensitivity and specificity. Accordingly, the analysis of the oral microbiota in our samples failed to detect the presence of *Streptococcus mitis* in patients with tumors localized in the oral cavity and hypopharynx, but not in the larynx.

Among the microorganisms responsible for the extrahepatic metabolism of alcohol to acetaldehyde, particularly in the oral cavity, *Candida albicans* possesses the ability to produce this mutagenic compound [24]. *Candida albicans* isolated from oral saliva samples from patients affected by potentially carcinogenic oral diseases produced dangerous amounts of acetaldehyde, as shown by an in vitro test [23]. Moreover, tobacco use together with alcohol consumption may favor adaptational changes, resulting in the upregulation of acetaldehyde formation by *Candida* spp. [23].

Drinking habits affected, as expected [13], the oxidative stress of the recruited individuals. Indeed, heavy drinkers had the highest antioxidant endogenous capabilities expressed as FORD, mainly when compared with social drinkers. Furthermore, social drinkers had low serum levels of ROS when measured in FORT. These data indicate that under stressful event conditions such as hospitalization for UADT carcinoma, oxidative stress and its endogenous counteraction system appear to be elevated in people who heavily drink and in non-drinkers. In particular, the antioxidant endogenous capacities were comparable between non-drinkers (as predictable) and heavy drinkers, showing that these people possess strong factors in terms of metabolizing alcohol.

Intriguingly, under the present experimental conditions, the presence or absence of oral acetaldehyde-producing bacteria affected, as expected, the oxidative stress of the enrolled individuals by increasing the FORD and FORT levels in those enrolled individuals showing oral acetaldehyde-metabolizing bacteria, thus evidencing a further link between UADT carcinomas, the oral microbiota, and the toxic effects of acetaldehyde [13]. However, the observed changes in oxidative stress did not influence the cancer district location and also the carcinoma severity (according to TMN) observed at both the first hospitalization and the 9-months follow-up. Furthermore, data on oxidative stress failed to detect significant correlations between drinking habits based on EtG and both carcinoma severity and location.

As for the study of ADH gene polymorphisms, we found that only the “CGTCGTCCC” haplotype was more frequent in the general population than in the enrolled individuals, but no effects of the haplotype on both alcohol habits and/or the presence in the mouth of the bacteria producing acetaldehyde were disclosed. Previous data show that the associations between alcohol dependence and polymorphisms in the alcohol-metabolizing enzymes, ADH and ALDH, are of great interest, because these variations can condition ethanol metabolism in humans. In particular, ADH gene polymorphisms have been the subject of extensive and continuing research [30,65,66], and the discovered allelic variations were found to produce effects on alcohol metabolism, even in vivo. It is known that alcohol metabolism already begins at the level of the mucosa of the upper digestive tract, and this evidence has been confirmed by the presence of ADH7 gene expression in these tissues [67]. As is known, ADH is the first enzyme involved in alcohol metabolism, and it works by oxidizing alcohol into acetaldehyde. ADH genes are clustered on chromosome 4q22–23 (5′-ADH7-ADH1C-ADH1BADH1A-ADH6-ADH4-ADH5-3′). Our haplotype analysis evidenced that the “CGTCGTCCC” type is more frequent in the general population than in carcinoma patients, suggesting a “protective role” for this haplotype against the damage induced by alcohol at the level of the UADT. Data on ADH7 and ALDH2 expression from peritumoral tissues evidenced an unbalanced level of ADH7. With the same expression of ALDH2, ADH7 results are more expressed in patients not carrying the “CGTCGTCCC” haplotype, which, consequently, may induce an accumulation in the acetaldehyde. Therefore, the supposed “protective role” could be explained in this way. Indeed, both ADH7 and ALDH2 crucially participate in the metabolism of acetaldehyde, and an imbalance in these enzymes might affect acetaldehyde degradation [68].

It is of interest to see in further studies if any correlation can be observed between the presence of the “CGTCGTCCC” haplotype and the severity of UADT carcinoma (in terms of histology), prognosis, and resistance to therapies. Moreover, as far as we know, there are no previous data in the literature about the relationship between the expression levels of the ADH7 and ALDH2 genes and the amount of the respective proteins produced in humans. This lack of knowledge could be another starting point for future studies.

The strength and novelty of this investigation were to classify the enrolled individuals according to both the alcohol consumption and the presence of bacteria producing acetaldehyde. The groups of patients were distributed not only according to criteria correlated to severe UADT carcinomas. In particular, the present retrospective study strongly focused at the level of (i) the EtG hair values and (ii) the oral occurrence of *Streptococcus mitis*, *Candida albicans*, *Candida glabrata*, and *Neisseria subflava*. In order to unveil the risk of alcohol consumption at the onset of UADT carcinoma, previous studies were mainly based on interviews that aimed to discover alcohol drinking habits without measuring the real, long-lasting alcohol consumption [69,70,71].

This investigation obviously has some limitations. The *n* of the carcinomas patients was different and small (but this also depends on the quite restricted inclusion/exclusion criteria of the study and because the findings of this retrospective report originate from a single university hospital during the recent pandemic), so some biases could have occurred. Additionally, the absence of longer clinical information from further follow-ups could be considered a study limitation. Moreover, the EtG analysis in the hair is reliable for heavy and social alcohol drinking, but for accidental alcohol exposure or very low drinking, the method might not fully disclose small EtG values. Indeed, an incident of accidental exposure in real life may involve ethanol in foods, mouthwash, inhalation of alcohol from topical use, and even over-the-counter medications containing ethanol. This topic is another limitation of our study.

## 5. Conclusions

The availability of new biomarkers is a very topical research field, especially for pathologies such as cancer. In our pilot study, albeit with a rather limited sample, the need to objectively estimate alcohol consumption is evident, in order to establish its role in the pathogenic process. Moreover, our findings suggest and consolidate the evidence that the effective carcinogenic role of alcohol depends on many factors beyond subjective consumption, not least the presence of genetic haplotypes, which is an interesting factor to monitor. Finally, the analysis of the oral cavity microbiota seems to provide further interesting information on the risk of generating carcinogenic metabolites locally. The strong limitation of this type of evaluation remains the fact that the analysis is conducted in patients with full-blown cancer, which is why it is not possible to define whether a dysbiosis is caused by the pathology, or whether it is an active component in the pathogenesis of UADT cancers. In future studies, it will be interesting to recruit patients who are screened for pre-cancerous lesions in order to identify innovative early cancer biomarkers.

## Figures and Tables

**Figure 1 antioxidants-12-01233-f001:**
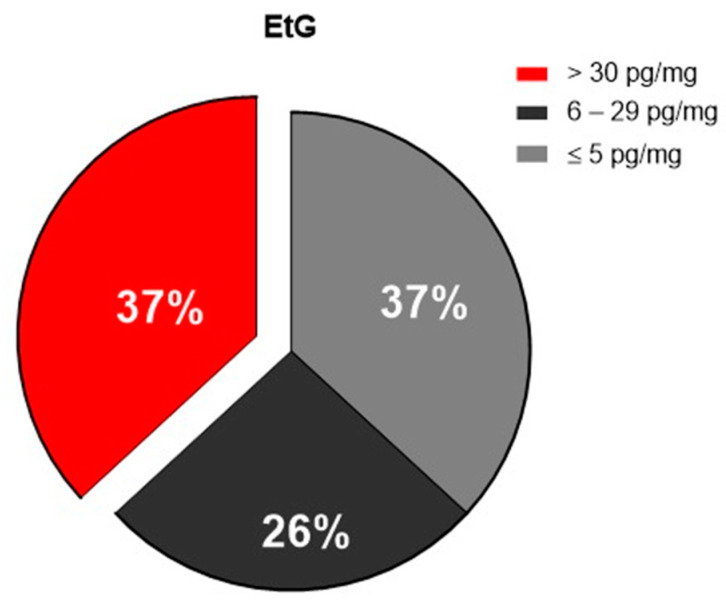
Pie chart representing the percentage of the enrolled patients, according to EtG hair values: heavy drinkers in red (EtG > 30 pg/mg), social consumers in black (6–29 pg/mg), and non-drinkers in gray ≤ 5 pg/mg.

**Figure 2 antioxidants-12-01233-f002:**
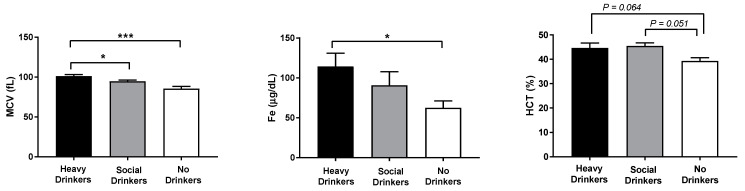
Mean corpuscular volume (MCV), iron (Fe), and hematocrit (HCT) data of the recruited individuals. The error bars indicate pooled standard error of the mean (SEM) derived from appropriate error mean square in the ANOVA. The asterisks (*** *p* < 0.01; * *p* < 0.05) indicate post hoc differences between groups.

**Figure 3 antioxidants-12-01233-f003:**
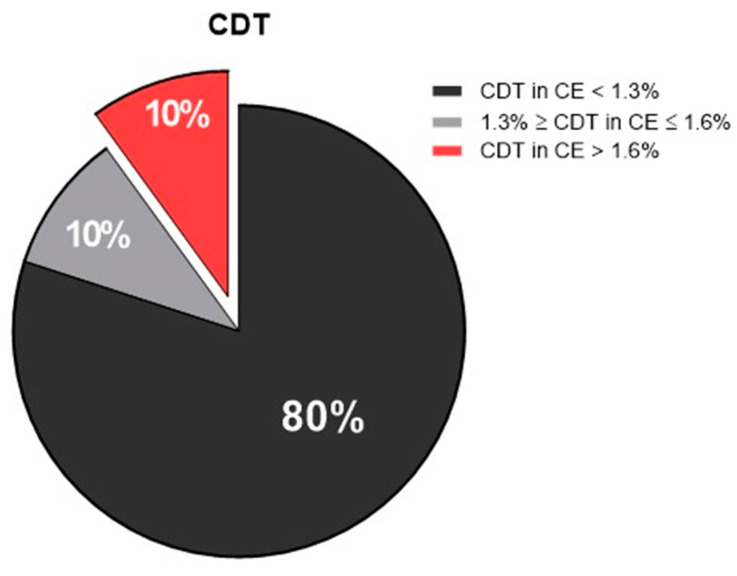
Pie chart showing the percentage of the enrolled patients, according to carbohydrate-deficient transferrin (CDT) values: non-drinkers in black (CDT in CE < 1.3%), social consumers in gray (1.3% ≥ CDT in CE ≤ 1.6%), and heavy drinkers in red (CDT in CE > 1.6%). CE, capillary electrophoresis.

**Figure 4 antioxidants-12-01233-f004:**
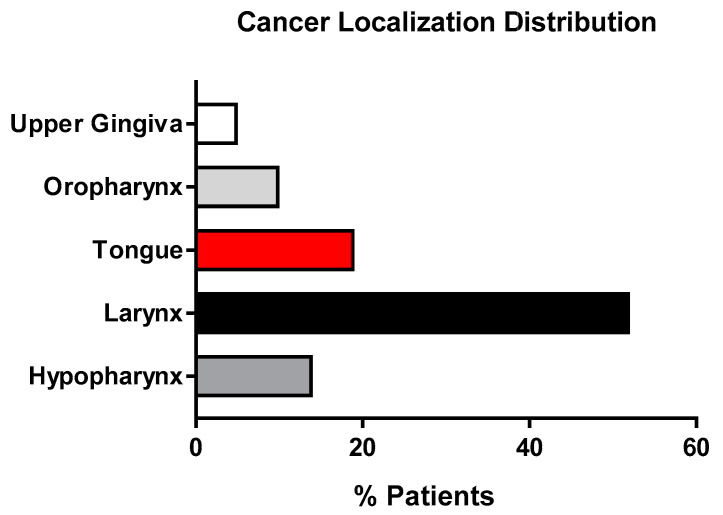
Histograms representing the cancer localization % distribution of the population enrolled in the study.

**Figure 5 antioxidants-12-01233-f005:**
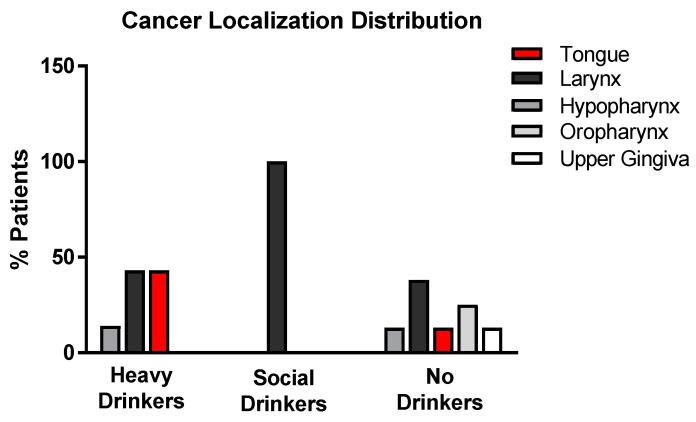
Histograms representing the cancer localization % distribution in the population enrolled in the study according to EtG values.

**Figure 6 antioxidants-12-01233-f006:**
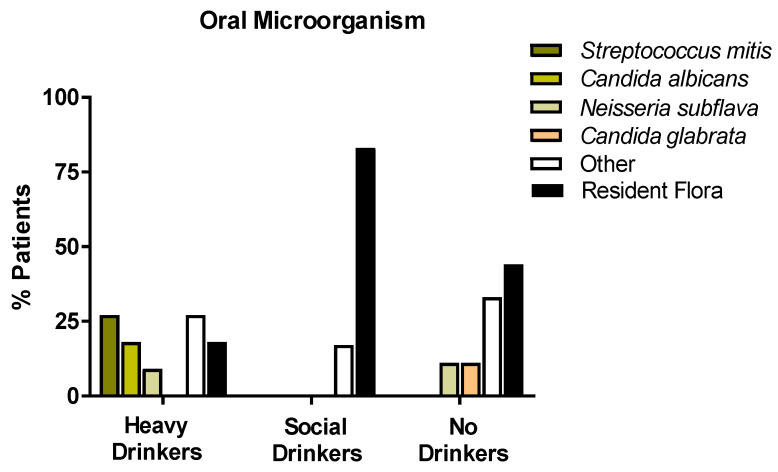
Histograms representing the oral microorganism % distribution of the population enrolled in the study, according to the classification of alcohol consumption (EtG values), with a particular attention for the metabolizing ethanol microorganisms (*Streptococcus mitis*, *Candida albicans*, *Candida glabrata*, *Neisseria subflava*).

**Figure 7 antioxidants-12-01233-f007:**
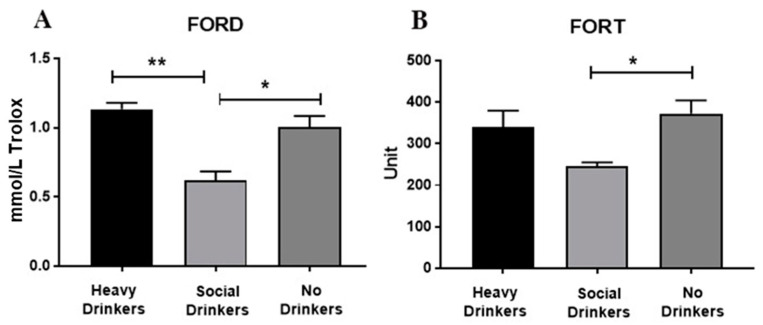
Serum FORD (panel (**A**)) and FORT (panel (**B**)) data of the recruited individuals. The error bars indicate pooled standard error of the mean (SEM) derived from appropriate error mean square in the ANOVA. The asterisks (** *p* < 0.01; * *p* < 0.05) indicate post hoc differences between groups.

**Figure 8 antioxidants-12-01233-f008:**
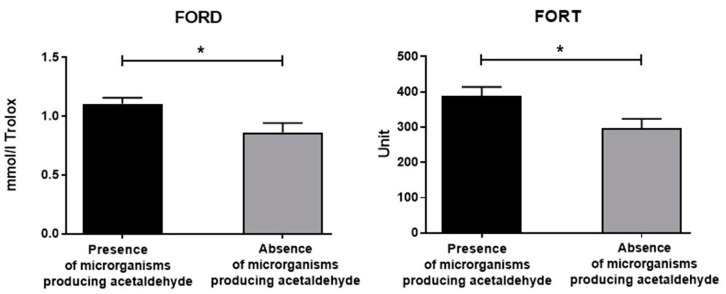
Serum FORD and FORT data of the recruited individuals with or without oral bacteria producing acetaldehyde. The error bars indicate pooled standard error of the mean (SEM) derived from appropriate error mean square in the ANOVA. The asterisks (* *p* < 0.05) indicate post hoc differences between groups.

**Figure 9 antioxidants-12-01233-f009:**
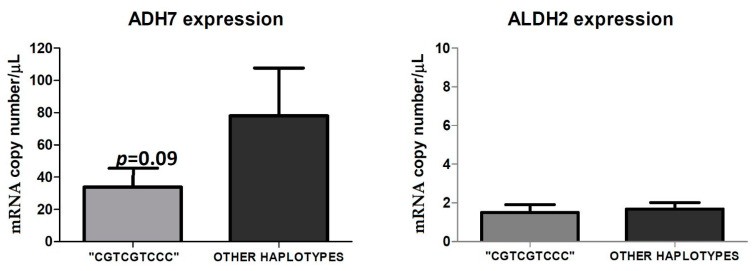
ADH7 and ALDH2 expression in the available peritumoral tissues in cancer patients with the haplotype “CGTCGTCCC” compared to cancer patients who have other haplotypes. The figure represents the mean of ADH7 and ALDH2 expression, in RNA copy number/µL, obtained by digital droplet PCR of cancer patients presenting with the haplotype “CGTCGTCCC” (*n* = 4) versus those with other haplotypes (*n* = 6).

**Table 1 antioxidants-12-01233-t001:** General characteristic of the enrolled patients: sex, age, cancer localization, and stadiation—clinical or pathological tumor-node-metastasis staging system (AJCC/UICC—TNM—8th Edition).

Patient ID#	Sex	Age (Years)	Cancer Localization	p/cTNM	Declared Smoking (n° Cigarettes Per Day/Years of Smoking)	9 Months Follow-Up
1	M	60	Hypopharynx	cT4bN3bM0	10/45	Alive with recurrent local carcinoma
2	F	57	Hypopharynx	pT4aN0M0	40/not declared	Alive, no recurrent local carcinoma
3	M	72	Larynx	pT4aN0M0	20 per day up to 1981	Alive, no recurrent local carcinoma
4	M	58	Hypopharynx	pT3N3bM0	40/40	Alive with severe recurrent local carcinoma
5	M	54	Larynx	pT3N3bM0	30/35	Alive, no recurrent local carcinoma
6	M	63	Larynx	pT4aN0M0	20/48	Alive, no recurrent local carcinoma
7	M	57	Tongue	pT3N1M0	10/13	Alive, no recurrent local carcinoma
8	M	49	Larynx	pT3N0M0	50/35	Alive, no recurrent local carcinoma
9	M	66	Oropharynx	cT4N2bM0	10/10	Alive with recurrent local carcinoma
10	F	58	Tongue	pT3N3bM0	20/20	Alive, no recurrent local carcinoma
11	M	59	Upper Gingiva	cT4bN2cM1	40/30	Deceased
12	M	60	Tongue	cT4aN2bM0	20/40	Alive with recurrent local carcinoma
13	M	53	Oropharynx	cT4aN3bM0	20/35	Alive with severe recurrent local carcinoma
14	M	46	Tongue	pT1N0M0	20/30	Alive, no recurrent local carcinoma
15	M	65	Larynx	pT3N0M0	80 per day up to 60 years	Alive with severe recurrent local carcinoma
16	M	70	Larynx	pT4aN3bM0	20/56	Deceased
17	F	75	Larynx	pT3N0M0	30/47	Alive, no recurrent local carcinoma
18	M	56	Larynx	pT3N0M0	40/10	Alive, no recurrent local carcinoma
19	M	71	Larynx	pT3N3bM0	25 cigarettes up to 2014	Alive with severe recurrent carcinoma
20	F	60	Larynx	pT2N0M0	20/35	Alive, no recurrent local carcinoma
21	M	67	Larynx	pT4aN0M0	20/15	Alive, no recurrent local carcinoma

**Table 2 antioxidants-12-01233-t002:** Common haplotypes from investigated polymorphism in the recruited patients and controls. χ^2^, chi-square; df, degree of freedom; *p*, *p* value; CI, confidence interval; and OR, odds ratio. * *p* values < 0.0007 (significant threshold after Bonferroni correction). N refers to the number of alleles.

* Haplotype	Ctrls (*n* = 40) N (Frequency)	Tumor (*n* = 42) N (Frequency)	χ^2^, df	*p*	OR	95% CI
C G T C G T C C C	10 (0.25)	3 (0.068)	5.295, 1	0.0214 *	4.556	1.153 to 18.00
C G T T A A C C G	1 (0.025)	6 (0.136)	3.402, 1	0.0651	0.1624	0.01865 to 1.414
C G C T A A C C G	10 (0.25)	11 (0.25)	0.0000, 1	1.0000	1.000	0.3719 to 2.689
C G C C G T C C C	9 (0.225)	17 (0.386)	2.553, 1	0.1101	0.4611	0.1768 to 1.203
C G C C G T G T C	3 (0.075)	2 (0.045)	0.3267, 1	0.5676	1.703	0.2695 to 10.76

## Data Availability

Data are available on request due to ethical reasons.

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
