# Peer review of "The Impact of Alcohol Consumption and Oral Microbiota on Upper Aerodigestive Tract Carcinomas: A Pilot Study"

_antioxidants, 2023, doi:10.3390/antiox12061233_

Round 1
Reviewer 1 Report
The authors have examined the oral microflora of 21 patients with upper aerodigestive tract carcinomas. The study was considered a pilot study and microflora, genetic and mRNA analyses were done. In addition to the limitations mentioned by the authors (small sample size, reverse causation concerning cancer flora) the definitions, analyses and structure of the manuscript have shortcomings. Examples include:
· The authors have used culture-based methods to identify bacteria of interest. In an explorative setting 16s rRNA sequencing would provide important information.
· Abstract: The abstract seems long (300 words) and the main findings are hard to spot.
· The term “Real alcohol” does not seem justified. Please rephrase, you have measured two markers of alcohol consumption.
· The abstract should state the method for analyzing bacterial composition.
· The sentence “Most of the abstemious or social drinkers had a prevalence of resident microflora” is difficult to understand. All individuals have a resident microflora.
· Line 85-86: what is meant by C albicans isolated from an oral pathogen? C albicans may be considered an oral pathogen. What pathogens do the authors mean? Please clarify.
· The lines 93-104 represent the aims of the study and is long, detailed and contains definitions and methodology. Please shorten.
· In the Methods section, the description of the patient cohort is important. Were patients enrolled at the time of diagnosis before treatment start? What was the time between diagnosis and sampling? The four lines concerning hair length 5 cm seem excessive.
· A 15 day washout after antibiotic therapy seems very short compared to other studies of GI microflora. Are there any studies that suggest that the recovery time is as low as 15 days?
· Line 125. It is stated that centrifugation was used to separate serum from plasma within the same Vacutainer. Is this possible at all? The purpose is normally to separate plasma or serum from blood cells.
· In the section describing EtG, please state the sensitivity and specificity for heavy drinking as well as the definition of heavy drinking. Please provide similar information about the performance of CDT.
· The classification of degrees of alcohol consumption into three groups is very vague, using alcohol units or grams/week or month is acceptable. Lack of definitions is a major weakness.
· Table 1 and line 316: the term “Cancer District” is uncommon, please change to correct term.
· Figure 2 presents results of analyses that were not mentioned in the methods section.
Please see above.
Author Response
The authors have examined the oral microflora of 21 patients with upper aerodigestive tract carcinomas. The study was considered a pilot study and microflora, genetic and mRNA analyses were done. In addition to the limitations mentioned by the authors (small sample size, reverse causation concerning cancer flora) the definitions, analyses and structure of the manuscript have shortcomings. Examples include:
- The authors have used culture-based methods to identify bacteria of interest. In an explorative setting 16s rRNA sequencing would provide important information.
Reply: we agree with the referee that the 16S analysis can offer a wider information to have an overview of the microbial composition of the oral cavity. However, this is beyond the scope of our work. In fact, our primary interest was to evaluate, in the oral cavity, the presence of specific microorganisms (Candida albicans, Candida glabrata, Neisseria subflava, and Streptococcus mitis) capable of locally metabolizing alcohol to acetaldehyde. The culture media analysis is, indeed, quite used to detect the microorganisms of interest in our study.
- Abstract: The abstract seems long (300 words) and the main findings are hard to spot.
Reply: as suggested, we have shortened and updated the abstract.
- The term “Real alcohol” does not seem justified. Please rephrase, you have measured two markers of alcohol consumption.
Reply: according to the observation of the reviewer, we rephrased the term “real alcohol”.
- The abstract should state the method for analyzing bacterial composition.
Reply: as suggested we have included the method for analyzing bacterial composition in the abstract (lines 28 of the revised paper).
The sentence “Most of the abstemious or social drinkers had a prevalence of resident microflora” is difficult to understand. All individuals have a resident microflora.
Reply: we agree that the sentence could be ambiguous. We removed the sentence from the abstract.
- Line 85-86: what is meant by C albicans isolated from an oral pathogen? C albicans may be considered an oral pathogen. What pathogens do the authors mean? Please clarify.
Reply: we do thank the reviewer for the observation. We clarified the sentence in the new version of the manuscript. (line 81 of the revised paper)
- The lines 93-104 represent the aims of the study and is long, detailed and contains definitions and methodology. Please shorten.
Reply: according to the referee we shortened this part (lines 89-96 of the revised paper)
- In the Methods section, the description of the patient cohort is important. Were patients enrolled at the time of diagnosis before treatment start? What was the time between diagnosis and sampling? The four lines concerning hair length 5 cm seem excessive.
Reply: according to the referee, we improved the description of the patient cohort and shortened the descriptive section concerning hair sampling (lines 111-115 of the revised paper).
- A 15 day washout after antibiotic therapy seems very short compared to other studies of GI microflora. Are there any studies that suggest that the recovery time is as low as 15 days?
Reply: To the best of our knowledge there are no studies in the literature reporting the time of oral microbiome restoration after antibiotic therapy. The concentration of almost all antibiotics decreases below the MIC values within 12 to 18 hours of administration, depending on the distribution of the drug into the tissues, and it is quite clear that not only clinical but also microbiological efficacy depends on drug levels
above the MIC. Furthermore, it has to be considered that oral microbiome differs from gut microbiome, since it has been reported that the oral microbiome can exhibit stability over periods of at least 3 months (Utter
DR et al. Front. Microbiol., 22 April 2016 Sec. Systems Microbiology Volume 7 - 2016 | https://doi.org/10.3389/fmicb.2016.00564 ), and the gut microbiome for up to 5 years (Faith JJ, Guruge JL, Charbonneau M, Subramanian S, Seedorf H, Goodman AL, et al. The long-term stability of the
human gut microbiota. Science. 2013;341:1237439.https://doi.org/10.1126/science.1237439.) witnessing substantial differences in the complexity of the composition. In different clinical settings i.e. bacteraemia, it is widely accepted that the isolation of the same germ 7-10 after the end of the antibiotic therapy is not to be
considered as a relapse/recurrence, but as a new infection. Therefore, it is our opinion that a previous antibiotic treatment lasting 15 days before sampling might have had a very marginal effect on oral microflora,
especially considering the bacterial species targeted in our study.
- Line 125. It is stated that centrifugation was used to separate serum from plasma within the same Vacutainer. Is this possible at all? The purpose is normally to separate plasma or serum from blood cells.
Reply: we do thank the reviewer for the observation. We clarified the sentence in the new version of the manuscript. (line 117 of the revised paper)
- In the section describing EtG, please state the sensitivity and specificity for heavy drinking as well as the definition of heavy drinking. Please provide similar information about the performance of CDT.
Reply: according to the referee we modified the section (lines 161-163 and 174-176 of the revised paper)
- The classification of degrees of alcohol consumption into three groups is very vague, using alcohol units or grams/week or month is acceptable. Lack of definitions is a major weakness.
Reply: The subdivision, in the work, into three large groups (no drinkers, social drinkers, and heavy drinkers) is based on the estimate of alcohol consumption obtained from the measurement of the EtG metabolite in the hair. Several pieces of evidence clearly show that the EtG in the hair provides info also on the long-lasting alcohol consumption. This is of primary interest in view of the potential contribution of alcohol consumption on the carcinogenic processes. It should be noted that people when asked, do not answer on the true degrees of alcohol consumption because of stigma feelings.
- Table 1 and line 316: the term “Cancer District” is uncommon, please change to correct term.
Reply: according to the referee we changed “Cancer District” with “Cancer Localization” (Tabel, Figures 4 and 5)
- Figure 2 presents results of analyses that were not mentioned in the methods section
Reply: we do thank the reviewer for the observation. We added the paragraph in the method section (lines 128-132 of the revised paper).
Reviewer 2 Report
Reviewer comments for antioxidants-2425143
The study investigated the correlation between alcohol consumption, oral microbiota, and upper aerodigestive tract (UADT) carcinomas among a cohort of patients, taking into account genetic factors and oxidative stress levels. This study sounds intriguing, informative, and well-written as well.
However, several minor issues exist that should be revised, details are as follows:
1. The author only uses cultured media to examine a limited number of commensal oral bacteria. However, it is important to note that not all oral commensal bacteria can be cultured using standard laboratory techniques. Therefore, it is recommended to employ 16S and 18S rDNA sequencing to assess the microbiota profile comprehensively.
2. The study lacks data from healthy control subjects. Including data from healthy individuals would provide a more conclusive understanding of the findings.
3. It is worth considering the significant physiological differences between males and females, including variances in hormones, commensal bacteria, metabolic enzymes, and other factors. Consequently, it would be beneficial to analyze the data separately for men and women to obtain more accurate insights.
Author Response
The study investigated the correlation between alcohol consumption, oral microbiota, and upper aerodigestive tract (UADT) carcinomas among a cohort of patients, taking into account genetic factors and oxidative stress levels. This study sounds intriguing, informative, and well-written as well.
Replay: we do thank the referee for his/her positive comments.
However, several minor issues exist that should be revised, details are as follows:
- The author only uses cultured media to examine a limited number of commensal oral bacteria. However, it is important to note that not all oral commensal bacteria can be cultured using standard laboratory techniques. Therefore, it is recommended to employ 16S and 18S rDNA sequencing to assess the microbiota profile comprehensively.
Reply: we agree with the referee that the 16S analysis can offer a wider information to have an overview of the microbial composition of the oral cavity. However, this is beyond the scope of our work. In fact, our primary interest was to evaluate, in the oral cavity, the presence of specific microorganisms (Candida albicans, Candida glabrata, Neisseria subflava, and Streptococcus mitis) capable of locally metabolizing alcohol to acetaldehyde. The culture media analysis is, indeed, quite used to detect the microorganisms of interest in our study.
- The study lacks data from healthy control subjects. Including data from healthy individuals would provide a more conclusive understanding of the findings.
Reply: the aim of the present study was to put on evidence, inside of a cohort of patients affected by UADT cancers, if and how alcohol consumption and the presence of acetaldehyde- forming bacteria could impact on the presence of such cancers. Actually, we do not believe that, in this observational clinical study, data on healthy subjects could provide information directly correlated with the purpose of the study.
- It is worth considering the significant physiological differences between males and females, including variances in hormones, commensal bacteria, metabolic enzymes, and other factors. Consequently, it would be beneficial to analyze the data separately for men and women to obtain more accurate insights.
Reply: in all the Statistical Analyses we carefully took into account the sex effect but since no relevant differences were found this factor was pulled out (lines 247-248).
Round 2
Reviewer 1 Report
The manuscript has been improved. I wish the authors good luck with larger studies!
Minor grammatical adjustments may be done when receiving proofs.
Reviewer 2 Report
All the comments have received appropriate replies.